# PGNet: Pipeline Guidance for Human Key-Point Detection

**DOI:** 10.3390/e22030369

**Published:** 2020-03-24

**Authors:** Feng Hong, Changhua Lu, Chun Liu, Ruru Liu, Weiwei Jiang, Wei Ju, Tao Wang

**Affiliations:** 1College of computer and Information, Hefei University of Technology, Hefei 230009, China; hfeng255@sina.cn (F.H.); jsdzlch@hfut.edu.cn (C.L.); cttjww@126.com (W.J.); juwei@mail.hfut.edu.cn (W.J.); wtustc@mail.ustc.edu.cn (T.W.); 2College of Electrical and Mechanical Engineering, Chizhou University, Chizhou 247000, China

**Keywords:** object detection, key-point detection, IoU, feature fusion

## Abstract

Human key-point detection is a challenging research field in computer vision. Convolutional neural models limit the number of parameters and mine the local structure, and have made great progress in significant target detection and key-point detection. However, the features extracted by shallow layers mainly contain a lack of semantic information, while the features extracted by deep layers contain rich semantic information but a lack of spatial information that results in information imbalance and feature extraction imbalance. With the complexity of the network structure and the increasing amount of computation, the balance between the time of communication and the time of calculation highlights the importance. Based on the improvement of hardware equipment, network operation time is greatly improved by optimizing the network structure and data operation methods. However, as the network structure becomes deeper and deeper, the communication consumption between networks also increases, and network computing capacity is optimized. In addition, communication overhead is also the focus of recent attention. We propose a novel network structure PGNet, which contains three parts: pipeline guidance strategy (PGS); Cross-Distance-IoU Loss (CIoU); and Cascaded Fusion Feature Model (CFFM).

## 1. Introduction

Deep-learning methods have been successfully applied to many fields, such as image recognition and analysis, speech recognition, and natural language processing, due to their automatic learning and continuous learning capabilities. Detection of human key points is a fundamental step in expounding human behavior, such as action analysis, action prediction, and behavior judgment. In addition, behavior prediction needs to capture the fine details of an object, such as video tracking and behavior prediction. A fast and effective key-point detection is of great practical value in predicting and tracking people’s behavior under special scenarios. 

Human key-point detection is a considerable undertaking in computer vision. Before 2014, researchers mainly solved the task by using SIFT, HOG, and other feature operators to extract features, and combined them with graph structure models to detect joint point positions. With the combination of deep learning and many tasks of computer vision achieving remarkable results, researchers have begun to try to combine it with human key-point detection tasks.

The main application of human body key-point detection is human body pose estimation. These methods involve detecting the location of human body key points and distinguishing artificially set key-point locations on the human body, separating human body key points from a given image. In [1], a novel method for the maintenance of temporal consistency is proposed, and maintained the temporal consistency of the video by the structured space learning and halfway temporal evaluation methods. Wang et al. [2] proposed a method for estimating 3D human poses from single images or video sequences. [3] explored the human action analysis in a specified situation, based on the human posture extraction by pose-estimation algorithm. Deep neural network (DNN) methods were used, composed of residual learning blocks for feature extraction and recurrent neural network for time-series data learning. However, although this method performs predictive analysis on the behavior of people in the video, using deep convolutional networks, the trade-offs in computational consumption and real-time performance are not fully considered, meanwhile showing that human pose estimation is an important research field of computer vision, and that human key-point detection is a front-end research of human pose estimation. In [4] it was illustrated that human body pose recognition is performed by comparing the shadow of the projection with the shadow of the human body under special circumstances, and proposed a normalization technique to bridge the gap and help the classifier better generalize with real data. Zhang et al. [5] proposed three effective training strategies, and exploited four useful postprocessing techniques and proposed a cascaded context mixer (CCM). [6] proposed an end-to-end architecture for joint 2D and 3D human pose estimation in natural images. However, the above uses deep convolutional networks for training and positioning. However, the down-sampling makes for a lack of spatial information at the deep level and a lack of semantic information at the shallow level. At the same time, the trade-off between calculation volume and efficiency also makes it difficult to consider performance of the network in terms of practicality. There are deficiencies in real-time and computational burden. Figure 1 below shows the detection results of the method proposed in this paper.

Substantial research has been done before in human key-point detection. The purpose of human key-point detection is to estimate the key points of a human body from pictures or videos; it is also an important link in some downstream applications prior to preprocessing, e.g., [4,7,8,9,10,11]. At present, convolutional neural networks show strong advantages in feature extraction. Various models have been proposed for features, as well as various evolutionary networks, some for extracting high-semantic information, and more attention to shallow spatial information. The structure of the model is also the focus of many scholars; coding-decoder, fusion mechanism, and feedback mechanism are responsible for the optimization and supplement of the network structure. [5] depicted a key-point graph network designed to extract object detection and object segmentation of key points. There was excellent performance, but easy overlap of key points when separating small objects. [6,12] proposed improved network mainly using anchor center points to detect small objects, but the efficiency of the whole network was reduced. In the feature extraction process, there are two main methods of feature extraction. One is box-of-free feature extraction [13,14,15], in which target detection is accomplished by embedding a cosine function or embedding a class of clusters in pixels. The other is based on frame-based feature extraction, but this method of embedding clusters has two major disadvantages in the extraction process [16]. One is that the global information of the picture cannot be fully considered, and the other is that the embedded information is mainly a cosine function, so there are many restrictions before embedding, and this method must be limited in the use process. Another feature extraction and positioning method is based on bounding box object detection. [13,14,17,18,19,20]. [13] addressed two limitations brought up by conventional anchor-based detection: (1) heuristic-guided feature selection; and (2) overlap-based anchor sampling. Specifically, an anchor-free branch is attached to each level of the feature pyramid, allowing box encoding and decoding in the anchor-free manner at an arbitrary level [14]. Han et al. proposed an efficient framework for real-time object tracking which is an end-to-end trained offline Fully Conventional Anchor-Free Siamese network; the network consists of correlation section, implemented by depth-wise cross correlation, and supervised section which has two branches, one for classification and the other for regression. [17] presented a monocular 3D object detection method with feature enhancement networks; 3D geometric features of RoI point clouds are further enhanced by the proposed point feature enhancement (PointFE) network, which can be served as an auxiliary module for an autonomous driving system.

The current popular framed object detection method is based on anchored framed feature extraction. This method maps the density of the anchored frame onto the feature heat map and further improves the border of the anchored image by predicting the offset. An important metric for framed object detection is intersection over union (IoU). [18] used the IoU of the union of the bounding boxes for multiple objects predicted by images taken at different times, termed mIOU, and the corresponding estimated number of vehicles to estimate the multi-level traffic status. [19] generated a tight oriented bounding box for elongated object detection which achieves a large margin of improvement for both detection and localization of elongated objects in images. [20] used multi-label classification as an auxiliary task to improve object detection, and the box-level features and the image-level features of multi-label are fused to improve accuracy. [21,22,23] demonstrated that the main problems of current IoU loss are the speed of convergence and the inaccuracy of iterative regression. Zhao et al. [24] proposed that Distance-IoU mainly predicts the target frame based on normalized data, which makes convergence speed of the network itself and the accuracy of feature extraction better, compared with the previous methods IoU and Genaralized-IoU. 

In this paper, we propose a novel network of human key-point detection. The main backbone of the network is Resnet50, in the way our model can accurately locate the key points of the human body; the model adopts the pipeline structure, which effectively optimizes communication and network computing before contradiction. By using the form of bus pipeline, the features extracted at each stage are recombined, so that efficiency and speed are greatly improved. With the optimized network, the features of each stage can be shared to a greater extent, and the contradiction between the semantic information of the shallow features and the spatial information of the deep features is solved. 

The improved PGNet network has excellent performance on the COCO datasets. We use the image-guided method to accurately extract the key points of the human body to complete the positioning, and consider the combination of the network structure features extracted by the shallow network and the semantic features extracted by the deep network. A good feature extraction actuator should contain two common features; one is spatial information and edge information with sufficient shallow features, and the extraction of such information is mainly done through multiple convolution and iterative convolution operations; the other is with abundant semantic information for more accurate localization to complete the classification. In addition, we use the cross-loss function in the design of the loss function, which performs well on the COCO dataset, and our main contributions are as follows: We introduce a kind of pipeline guiding strategy (PGS) to share the extracted features to all layers (shallow layers and deep layers) in the form of a pipeline. This allows each layer to better separate the background noise, and at the same time share the weight of the opposite transfer between each other.We propose a cross-fusion feature extraction mode. Combining this model with PGS enables shallow spatial information and deep semantic information to be combined through a pipelined bus strategy, so that the computational efficiency and the network’s separation of foreground and background can effectively remove the foreground noise and the background effective information at the edges is fully considered.We developed a crossed Distance-IoU loss function. To obtain the region of interest, we calculated the convergence and speed of the border regression. The Cross-Distance-IoU loss function used is based on the distance between the center points and the overlap area, and shows excellent results in the rectangular anchor border regression. The pipeline is used to guide the network to use the Distance-IoU loss function and the backbone network uses the GIoU loss function.

Table 1 shows that the backbone structure of the [12] network is the same, but due to the different processing methods of subsequent decoding fusion, the performance on the COCO dataset is different. In the case of the same encoding method, this paper uses deep convolution and 1*1 convolution, so that a trade-off between calculation volume and speed is satisfied in feature extraction. The proposed method improves the accuracy of the COCO dataset by 0.2% over the previous method. Table 1 demonstrates that our algorithm makes full use of the pipeline guidance method, and the accuracy of the COCO dataset exceeds the previous advanced algorithms.

## 2. Materials and Methods

In this section, we mainly introduce some studies related to this article, including the recent key-point detection method, the characteristics of the pipeline structure and the working principle and the loss function of border regress-IoU.

### 2.1. Key-Point Detection Method

Previous methods mainly optimized network structure improvement and used deeper network structures. However, these methods achieved satisfactory results in key-point detection. [6,12,24,25,26,27,28,29,30].

### 2.2. Pipeline Guidance Strategy

As the layers of the network become deeper and deeper, the joint parallel computing of multiple GPUs provides the possibility of speeding up the network. Multi-GPU is divided into multiple stages of the network, and the convolution operations and rectangular transformations of the network are performed in parallel, and communication between various operations is performed. The guidance mechanism is to use its own characteristics to supervise and complete the further optimization of its own information feature extraction. For example, shallow rich-edge information is used to guide the deep layer to better extract deep semantic information, feedback the deep semantic information to supervise the extraction of shallow edge information, and ultimately complete the performance optimization of the network structure, so that the receptive fields of different layers can play to their own advantages [31,32,33,34].

### 2.3. IoU Loss

In deep convolutional neural networks, the mainstream method in the process of feature extraction is the frame regression method. An important index for measuring this method is the loss function. The function of the loss function is to predict the distance between the target frame and the prediction frame. Current popular networks such as YOLOv3, SSD, and Faster R-CNN use GIoU, CIoU, or some improved loss functions combined with them [22,23]. Figure 2 shows that the number of positive bounding boxes after the NMS, grouped by their IoU with the matched ground truth.

IoU is an important indicator for neural networks to measure between ground truths and predicted images. In object detection of a bounding box, the object being detected is the minimum value of the rectangular border through multiple iterations.
(1)$$IoU=\frac{B\cap B^{gt}}{B\cup B^{gt}}$$
$B^{gt}$, B respectively ground truth and predicted images. 

Table 2 shows IoU operation logic, realized target point detection, and key fixed positioning through the same iterative operation multiple times. Despite the detection network frameworks being different, the regression calculation logic for predicting the borders to locate the borders in the target object is the same. Shengkai et al. [36] propose IoU-balanced loss functions that consist of IoU-balanced classification loss and IoU-balanced localization loss to solve poor localization accuracy, and this is harmful for accurate localization. [37] proposed visible IoU to explicitly incorporate the visible ratio in selecting samples, which included a box regressor to separately predict the moving direction of training samples. [23] Yan et al. proposed a novel IoU-Adaptive Deformable R-CNN framework for multi-class object detection, i.e., IoU-guided detection framework to reduce the loss of small-object information during training. Zheng et al. [38] proposed a Distance-IoU (DIoU) loss by incorporating the normalized distance between the predicted box and the target box, which converges much faster in training than IoU and GIoU losses.

## 3. Results

Based on the above, our model solves the accuracy and efficiency of key points in positioning, and optimizes the communication consumption due to many iterative operations and convolution operations. In the process of extracting features, the feature fusion mechanism is used to combine high-latitude semantic information with low-latitude spatial information, which makes for great efficiency in the process of locating key points of the human body. Figure 1 shown our proposed framework, which consists of three parts, in three branches—ResNet-51 is selected as the backbone network for picture feature extraction; there is adaptive strategy using pipeline guidance; and a cascaded feature fusion model. The framework of the network is shown in Figure 3.

### 3.1. Cascaded Fusion Feature Model

The main task of the Cascaded Fusion Feature Model (CFFM) is to extract the multi-layer features of the input picture and generate regions where key points are located. The traditional method is to directly use the multi-layer features to generate the prediction anchor frame and compare the ground truth picture to generate the key-point coordinates.

We propose the cascade fusion feature model to extract high-level features and low-level features; the high layers are rich in semantic correlation information and lack low-level spatial information. In contrast, the low layers are rich in edge and spatial features and lack semantic information. In particular, we build CFFM on ResNet-50, which will extract its features using conv1–5 layers. Considering the shallow layers simultaneously use a lot of computing resources, there is no significant improvement in performance, lack of edge, and spatial information during deep feature processes. We use the middle three layers to avoid the consumption of a large amount of spatial information during convolution calculations.

### 3.2. Pipeline Guidance Strategies

The process of locating key points of the human body is mainly done by analyzing human characteristics, locate key parts, and prepare for downstream video surveillance. We proposed combined traditional data parallelism with model parallelism enhanced with pipelining [39]. Through the structure of the pipelining, separate processing is performed on feature extraction and feature guidance, which effectively saves the resource consumption of the network structure in the communication process.

Pipeline-parallel training partitions the layers of the object being trained into multiple stages. Each layer contains a continuous set of structures in the model, as shown in Figure 3. The pipeline-type structure is used to guide the feature extraction at each stage. After the feature extraction, a convolution operation is used to fuse the features of the two branches on the pipeline and after the feature extraction to complete the key points. Figure 2 shows a network structure based on pipeline guidance, and Figure 3 is a diagram of key points of the human body using the PGNet network. [39]. Figure 4 shows that an example pipeline-parallel assignment with four machines and an example timeline at one of machines.

Because IoU loss can only be effective when the bounding box is reattached during the training process, there are steps where gradient optimization cannot be performed without any coincidence [36]. To overcome the disadvantage that the border boxes must have coincidence to take advantage of IoU losses, GIoU was proposed. Both these losses can make key-point detectors more powerful for accurate localization. According to Equation (1), IoU loss can be defined such that
(2)$$\;\;L_{IoU}=1-\frac{B\cap B^{gt}}{B\cup B^{gt}}$$

According to Equation (2), it can be known that the calculation of $L_{IoU}$ must be performed in an iterative manner only if there is intersection between the predicted target and the ground truth. GIoU was proposed to improve the gradient descent prediction operation of two bounding boxes without intersection. GIoU defines a distance, between which two bounding boxes can exist without crossing. GIoU is defined as such that
(3)$$GIoU=IoU-\frac{C-B\cup B^{gt}}{\left|C\right|}$$
where C is the smallest box covering B and B^gt^. Due to the introduction of the penalty term, the predicted box will move towards the target box in non-overlapping cases [40]. From Equation (3), we get $L_{GIoU}$ Loss such that
(4)$$\;L_{GIoU}=1-\mathrm{IoU}+\frac{\left|C-B\cup B^{gt}\right|}{\left|C\right|}$$

$L_{GIoU}$ loss aims to reduce the distance between the center point of the predicted box and the real box.

The cross-distance loss functions we propose inherit some of their inherent properties and are defined as
(5)$$L_{CDIoU}=1-\mathrm{IoU}+\frac{\rho^{2}\left(\delta ,\delta^{gt}\right)}{D^{2}}$$

Where, in Equation (5), δ, $\delta^{gt}$ denote the central points of B and B^gt^, ρ is the Euclidean distance, and D is the distance of the B and B_g_. Figure 5 shows that $\;L_{CDIoU}$ Distribution of bounding boxes for iterative training.

## 4. Discussion

### 4.1. Datasets and Evaluation Metrics

Our method of evaluating our designed network is on the COCO-2017 database, which is a large image dataset designed for object detection, segmentation, human key-point detection, thing segmentation, and subtitle generation.

In addition, the average precision (AP) metric is used to measure and evaluate the performance of PGNet. To illustrate the performance between the key-point location of the detection object and the key-point of the ground truth object, the results show that the method performs well.

### 4.2. Ablation Studies

The ablation experiment uses different backbone networks to regularize the method separately and unreasonably, and experiments on the network structure of this problem are based on six indicators. The benchmark database of the experiment is COCO val-2017. The experimental results shown below are obtained. The experimental results show that the network structure proposed in this paper is superior to other network structures in performance, as shown in Table 3.

Another part of the ablation experiment is to compare the results of the Eproch training using a pipelined structure. On object detection and image classification with small mini-batch sizes, CBN is found to outperform the original batch normalization and a direct calculation of statistics over previous iterations without the proposed compensation technique [41] in COCO val-2017. Figure 6 shows the training and test results.

## 5. Conclusions

In this paper, we propose an up-to-date type of human body key-point positioning network structure Piple-Guidance NeT. Considering that different layers contain incomprehensible features, the use of pipelined guidance in the structure allows the network to achieve a balance between the convolution calculations and the communication time between the layers, which improves the training speed of the network. In addition, the Cross-Distance-IoU mode is used in the training process, and the results are pleasing in different network backbones. Finally, regarding the COCO2017 dataset, the effectiveness of the algorithm is measured by the six parameters of the AP, and the effects demonstrate that the algorithm performs well. Compared with the current most advanced algorithms, the method improves the accuracy by 0.2%.

## Figures and Tables

**Figure 1 entropy-22-00369-f001:**
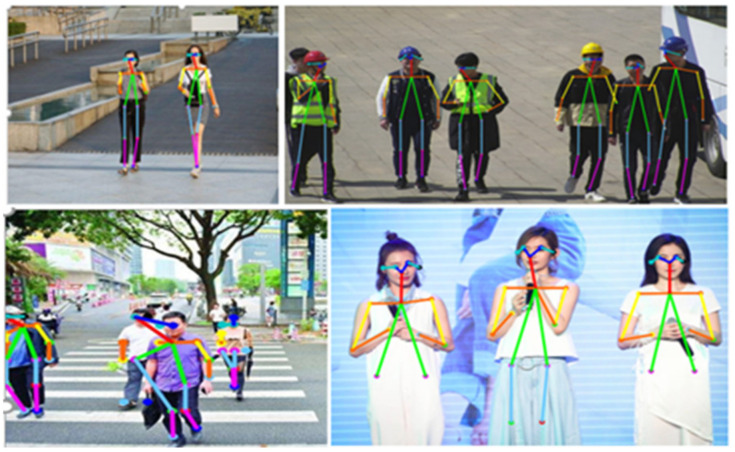
The proposed network to find key points of the human body.

**Figure 2 entropy-22-00369-f002:**
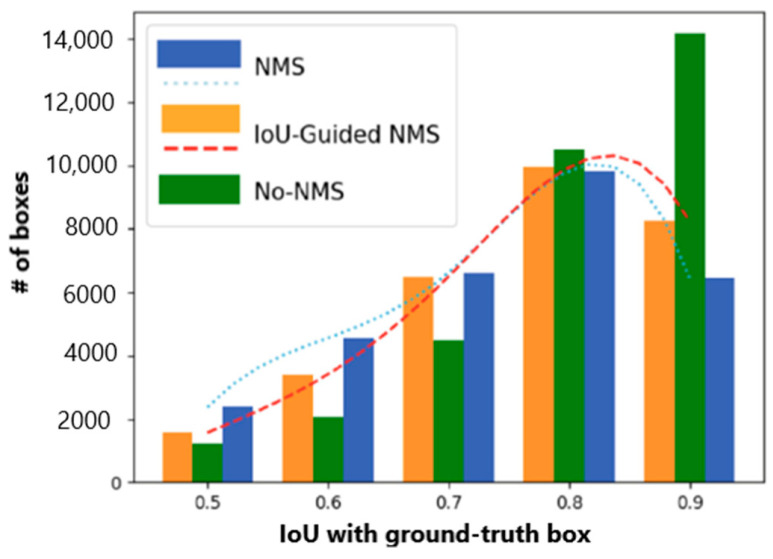
The number of positive bounding boxes after the NMS, grouped by their IoU with the matched ground truth. In traditional NMS (blue bar), a significant portion of accurately localized bounding boxes get mistakenly suppressed due to the misalignment of classification confidence and localization accuracy, while IoU-guided NMS (yellow bar) preserves more accurately localized bounding boxes [35].

**Figure 3 entropy-22-00369-f003:**
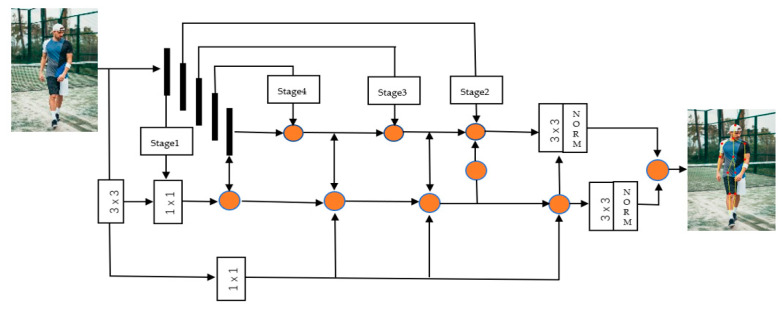
An of overview of proposed PGNet.ResNet-50 is used as the backbone. Using the cascaded fusion feature model (CFFM), the backbone network is divided into 5 stages, and the feature-guided network after the image is convolved is used to extract key-point features.

**Figure 4 entropy-22-00369-f004:**
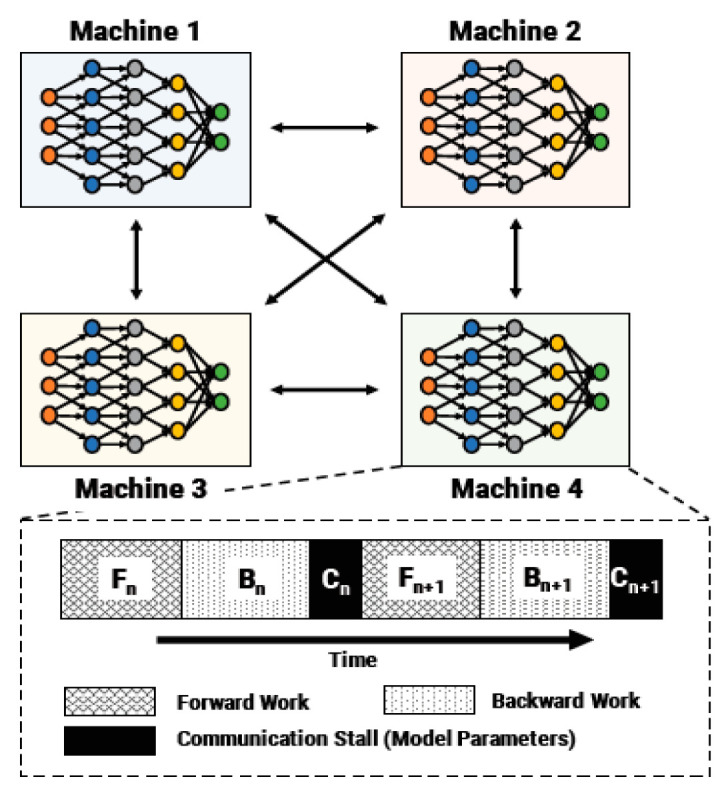
An example pipeline-parallel assignment with four machines and an example timeline at one of machines, highlighting the temporal overlap of computation and activation/gradient communication.

**Figure 5 entropy-22-00369-f005:**
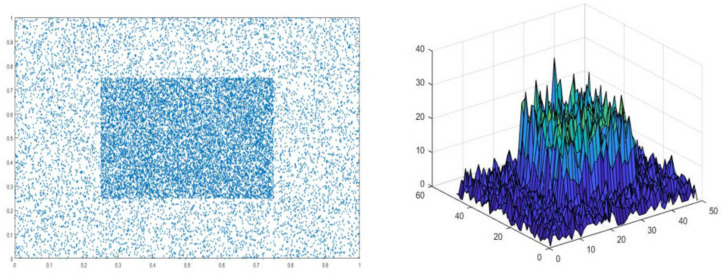
$L_{CDIoU}$ Distribution of bounding boxes for iterative training.

**Figure 6 entropy-22-00369-f006:**
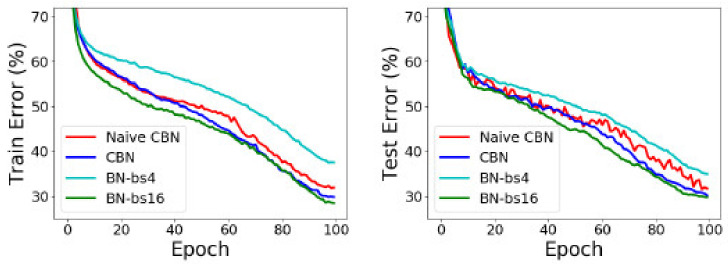
Comparison of epoch trained by this method and epoch of other training methods.

**Table 1 entropy-22-00369-t001:** Performance comparison of various network structures.

Method	Backbone	Decoder	Postprocessing	Performance
Mask-R-CNN [25]	ResNet-50-FPN	conv+deconv	offset regression	63.1AP@COCO
DHN [26]	ResNet-152	deconv	Flip/sub-pixel shift	73.7 AP@COCO
CNN [27]	VGG-19	conv	Flip/sub-pixel shift	61.8 AP@COCO
PGNN [28]	ResNet-50	GlobalNet	Flip/sub-pixel shift	68.7 AP@COCO
DetNet [29]	ResNet-50	deconv	Flip/sub-pixel shift	69.7 AP@COCO
DENSENETS [30]	ResNet-50	deconv	-	61.8AP@COCO
LCR-Net++ [12]	ResNet-50	deconv	Flip/sub-pixel shift	73.2AP@COCO
HRNet [6]	HRNet-152	1×1conv	Flip/sub-pixel shift	77.0AP@COCO
[5]	ResNet-101	deconv	Flip/sub-pixel shift	69.9 AP@COCO
PFAN [24]	VGG-19	multi-stage CNN	Flip/sub-pixel shift	70.2 AP@COCO
Proposed method	ResNet-50-Pipeline	Deconv+1×1conv	offset regression	77.2AP@COCO

**Table 2 entropy-22-00369-t002:** Logic operation based on bounding box regression.

**Alogrithm1 IoU** for two axis-Aligned BBox.
**Require:** -Corners of the bounding boxes:
$A_{1}$($x_{1}$,$y_{1}$),$B_{1}$($x_{2}$,$\;y_{1}$),$C_{1}$($x_{2}$,$\;y_{2}$),$D_{1}$($x_{1}$,$\;y_{2}$),
$A_{2}$ ($x_{1}^{’}$,$y_{1}^{’}$),$B_{2}$($x_{2}^{’}$,$y_{1}^{’}$),$\;C_{2}$($x_{2}^{’}$,$y_{2}^{’}$),$\;D_{2}$($x_{1}^{’}$,$y_{2}^{’}$),
Where $x_{1}$≤$x_{2}$,$y_{2}$≤$y_{1}$,and $x_{1}^{’}\leq x_{2}^{’}$,$\;y_{2}^{’}\leq y_{1}^{’}$
Ensure: - IoU value;
1:▲The area of $B_{g}:$ $Area_{g}=\left(x_{1}-x_{2}\right)\times \left(y_{1}-y_{2}\right);$
2:▲The area of $B_{d}:$ $Area_{d}=\left(x_{2}^{’}-x_{1}^{’}\right)\times \left(y_{1}^{’}-y_{2}^{’}\right);$
3:▲The area of overlap:$Area_{overlap}=(\max\left(x_{2},x_{2}^{’}\right)-$
min($x_{1},x_{1}^{’}$))$\times \left(\max\left(y_{1},y_{1}^{’}\right)-\min\left(y_{2},y_{2}^{’}\right)\right)$;
4:▲$IOU=\frac{Area_{ocerlap}}{Area_{g}+Area_{d}-Area_{overlap}}$**;**

**Table 3 entropy-22-00369-t003:** Parameter comparison of various network structures after different regularization processing.

Backbone	Norm	AP^bbox^	AP_50_^bbox^	AP_75_^bbox^	AP_S_^bbox^	AP_M_^bbox^	AP_L_^bbox^
ResNet50+FPN	GN	37.8	59.0	40.8	22.3	41.2	48.4
	syncGN	37.7	58.5	41.1	22.3	40.2	48.9
	CBN	37.8	59.8	40.3	22.5	40.5	49.1
ResNet101+FPN	GN	39.3	60.6	42.7	22.5	42.5	48.8
	syncGN	39.3	59.8	43.0	22.3	42.9	51.6
	CBN	39.2	60.0	42.2	22.3	42.6	51.8
ResNet50+proposed	GN	39.3	60.7	42.6	22.5	43.2	48.1
	syncGN	39.3	59.8	43.5	23.4	43.7	51.9
	CBN	39.4	59.8	43.2	23.1	42.9	52.6

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
