# Peer review of "PGNet: Pipeline Guidance for Human Key-Point Detection"

_entropy, 2020, doi:10.3390/e22030369_

Round 1

Reviewer 1 Report

1. The structure of the introduction part does not follow the general framework of most technical papers. http://abacus.bates.edu/~ganderso/biology/resources/writing/HTWsections.html#introduction 2. Presentation of EVERY equation is wrong and not fluent in terms of logic. Rewrite them in a way can be found in the classic textbook on Linear Algebra at http://linear.axler.net/LinearAbridged.pdf or your own textbook on Calculus. 3. 5. Conclusions-> 5. Conclusion Conclusion is a part of paper describing some conclusions. 4. Every figure is lossy. They should be re-exported with command export_fig('x.png','-transparent','-m2') http://www.mathworks.com/matlabcentral/fileexchange/23629-export-fig 5. Every author should try to make the manuscript can be read through with a good mood.

Author Response

Reviewer:

Sincerely thank you for your review of my article. I have responded to your comments in the attachment。

     thank you!

Reviewer 2 Report

Thank in advance for the confidence to be a reviewer. In my own opinion, the objective in this article is good, but there are many errors that must be corrected before being published.
The following comments address only aspects of formatting, without thoroughly reviewing the content, but that detract from the quality of the article at first sight.

  1. I would like to see a discussion of the research carried out in the field in order to see the novelty of this research.
  2. On line 50 there are 9 references, but not discussed.
  3. Line 55. Type errors.
  4. The paragraphs are very long and the central idea is lost. In addition, many ideas appear in the same paragraph. They must be separated to understand in a better way.
  5. There are references that I do not known to which paragraph they belong. Points must appear in certain sentences. 
  6. After a reference there must be a point…. And in this version it does not exist.
  7. What is the main research problem? Please discuss prior research in that area. 
  8. Existing literature should be discussed to indicate where the contribution of your research is. The literature discussion is not observed.
  9. Many of the references appear in one place. For example, on line 118 there are 8 references, on line 131 there are 4.
  10. Table 1 is called after it simply appears. Please call first the tables and then show them. 
  11. I would liked to see a discussion regarding the different methods set out in Table 1, so I could see the GAP in which you want to contribute.
  12. I believe that a native English speaker must support the translation process and final editing before sending it to another journal.
  13. Table 2 never is called.
  14. There are many paragraphs with different line spacing. It is recommended to use the magazine's own template or be consistent.

Author Response

Dear Reviewer:

 Sincerely thank you for your review of my article. I have responded to your comments in the attachment。

     Thanks you!

Round 2

Reviewer 1 Report

  1. '、' is a Chinese punctuation.
  2. Embed every equation among a text sentence.
    Delete "as follows:" before every equation.
    As for how to present an equation, refer to 
  3. Every figure is seriously lossy.
    As for Fig. 3, redraw it in Visio and embed as an object.
    As for other figures, re-export from Matlab .emf format.
    https://wiki.fileformat.com/image/emf/

  4. Check tense of every sentence by referring to 
    http://cc.oulu.fi/~smac/TRW/tense_handout.htm

Author Response

Point 1: '、' is a Chinese punctuation.

Response 1: According to the reviewer’s suggestion, we make modifications in the full text “、” section.

Point 2: Embed every equation among a text sentence. Delete "as follows:" before every equation. As for how to present an equation, refer to http://linear.axler.netLinearAbridg-ed.pdf

Response 2: According to the reviewer’s suggestion,we make modifications in the embed every equation.

Point 3: Every figure is seriously lossy. As for Fig. 3, redraw it in Visio and embed as an object.As for other figures, re-export from Matlab .emf format.https://wiki.fileformat.com-imag/emf/

Response 3: Considering the reviewer’s suggestion, all Figs have been revised.

Point 4: Check tense of every sentence by referring to http://cc.oulu.fi/~smac/TRW/ten-se_handout.htm

Response 4: According to the reviewer’s suggestion, the manuscript is revised by an English native speaker, by referring to http://cc.oulu.fi/~smac/TRW/ten-se_handout.htm.

Reviewer 2 Report

There are too much formatting problems. For example, the paragraph on line 35 to 39 has a different line spacing.
The style of references has not been corrected. For example, in reference 1 I  do not know to which paragraph to assign it. Reference 2, the correct thing is to put Liu et al. [2] proposed .... This problem is repeated in other references.
References 13, 14, 17-20 are not discussed and it is not known to which paragraph they should be assigned.
Line 143, words as Table 1 is repeated.
Reference {Rogez, 2019 # 183} is not presented adequately.

My last comments regarding the paper contribution is now clear, however, there is still a lot work to do with this paper to be accepted for published.

Author Response

Point 1: The style of references has not been corrected. For example, in reference 1 I  do not know to which paragraph to assign it. Reference 2, the correct thing is to put Liu et al. [2] proposed .... This problem is repeated in other references.
References 13, 14, 17-20 are not discussed and it is not known to which paragraph they should be assigned.
Line 143, words as Table 1 is repeated.
Reference {Rogez, 2019 # 183} is not presented adequately.

Response 1: According to the reviewer’s suggestion, we make modifications in the “The style of references” section. For example, in reference 1 paragraph to assign it in Line 42.

Reference 2, we make modifications in the Line 45. This problem is repeated in other references which has been solved.

According to the reviewer’s suggestion, References 13, 14, 17-20 hava been discussed .

According to the reviewer’s suggestion, words as Table 1 is repeated and the overage has deleted.

According to the reviewer’s suggestion,reference {Rogez, 2019 # 183} is not presented adequately,we make modifications in the Line 143.
